# Evaluation of Precision Livestock Technology and Human Scoring of Nursery Pigs in a Controlled Immune Challenge Experiment

**DOI:** 10.3390/ani13020246

**Published:** 2023-01-10

**Authors:** Eduarda M. Bortoluzzi, Mikayla J. Goering, Sara J. Ochoa, Aaron J. Holliday, Jared M. Mumm, Catherine E. Nelson, Hui Wu, Benny E. Mote, Eric T. Psota, Ty B. Schmidt, Majid Jaberi-Douraki, Lindsey E. Hulbert

**Affiliations:** 1Department of Animal Sciences and Industry, Kansas State University, Manhattan, KS 66506, USA; 2Department of Animal Science, University of Nebraska-Lincoln, Lincoln, NE 68505, USA; 3Department of Statistics, Kansas State University, Manhattan, KS 66506, USA; 4Department of Electrical and Computer Engineering, University of Nebraska-Lincoln, Lincoln, NE 68588, USA; 5Department of Mathematics, Kansas State University, Manhattan, KS 66506, USA; 61-DATA, Kansas State University Olathe, Olathe, KS 66061, USA

**Keywords:** machine learning, Suidae domesticus, ethology

## Abstract

**Simple Summary:**

One of the most challenging setbacks for intensive swine industries to improve animal welfare and sustainability is skilled labor. Human observations are needed to accurately identify sick or injured pigs. Still, animal caretakers are limited in the amount of time they take and the frequency of observations they can complete in a day. Here, using a controlled immune challenge in nursery pigs, a visual-based precision livestock technology can identify pigs with greater specificity and sensitivity than trained human observers. This system can monitor pigs 24 h a day, seven days a week, in group housing. The potential impact of this research may improve the laborer’s ability to treat animals at the individual level rather than the group level.

**Abstract:**

The objectives were to determine the sensitivity, specificity, and cutoff values of a visual-based precision livestock technology (NU*track*), and determine the sensitivity and specificity of sickness score data collected with the live observation by trained human observers. At weaning, pigs (n = 192; gilts and barrows) were randomly assigned to one of twelve pens (16/pen) and treatments were randomly assigned to pens. Sham-pen pigs all received subcutaneous saline (3 mL). For LPS-pen pigs, all pigs received subcutaneous lipopolysaccharide (LPS; 300 μg/kg BW; E. coli O111:B4; in 3 mL of saline). For the last treatment, eight pigs were randomly assigned to receive LPS, and the other eight were sham (same methods as above; half-and-half pens). Human data from the day of the challenge presented high true positive and low false positive rates (88.5% sensitivity; 85.4% specificity; 0.871 Area Under Curve, AUC), however, these values declined when half-and-half pigs were scored (75% sensitivity; 65.5% specificity; 0.703 AUC). Precision technology measures had excellent AUC, sensitivity, and specificity for the first 72 h after treatment and AUC values were >0.970, regardless of pen treatment. These results indicate that precision technology has a greater potential for identifying pigs during a natural infectious disease event than trained professionals using timepoint sampling.

## 1. Introduction

In livestock production, caretakers are responsible for accurately identifying, documenting, and caring for animals with illness or injury through single timepoint pen observation by assessing behavior or other physical problems (e.g., tail bites). Continuous, temporal, and spatial data of animal behavior are not yet accessible to producers, although scientists have long established that continuous behavior collection greatly enhances the sensitivity and specificity identification of sick or injured animals.

The nursery phase brings numerous challenges to the health and welfare of newly weaned pigs. Weaning is stressful because of pig-sow separation, exposure to a novel environment, transportation, handling, and comingling with unfamiliar pigs from different litters [1,2,3]. The stressors that newly weaned pigs experience disrupt homeostasis, affecting health status [4,5]. Psychosocial stressors from weaning are also potent activators of the stress axis, culminating in a complex hormone and immune response which affects their already naïve immune system [6]. The disruption of homeostasis, and the attempt to regulate the concentration of stress hormones, can impede immune regulation and subsequently increase the risks of morbidity and mortality [5].

The causes and incidences of morbidity and mortality in postweaning pigs have not been widely reported in scientific literature. However, postweaning mortality has been defined as having a complex multifactorial causation [7], with rates ranging from 5.6 to 7.6% for pigs entering the wean-to-finish phase and 3.6% for pigs entering the nursery phase [8,9]. Recently a study using 1316 cohorts of pigs, demonstrated a geometric average mortality of 8.69% for pigs entering the wean-to-finish phase [10]. Respiratory and gastrointestinal disorders are leading causes of morbidity and mortality in post-weaning pigs [7,11,12]. Early identification, segregation, and treatment of compromised pigs can decrease morbidity and prevent mortality. Identifying sick or injured pigs can be challenging when pigs are housed in groups, especially if individual pigs are in subclinical stages of pathology. Swine producers rely on animal technicians to identify pigs requiring attention [13]. Typically, animal technicians use scan sampling (scanning pens for individuals with sickness or injury signs), and this timepoint sampling is limited to once or twice a day. Experience and skills training are required to identify subtle abnormal behaviors correctly [14,15].

Moreover, visual observation by a caretaker is only a short snapshot in time that does not translate into overall daily pig behavior and is prone to human error and bias. The limitation of human observation may be overcome with advancements in precision livestock technology that can identify compromised pigs via audio, video, or wearable devices [16,17,18]. The NU*track* system was developed as a visual-based precision livestock technology platform for the identification, activity, and behavior tracking of group-housed pigs at the animal level [19,20,21,22]. Therefore, the first objectives of this current work were to establish the sensitivity, specificity, and area under the curve (AUC) of skilled technicians to correctly identify and distinguish experimentally induced sick pigs (lipopolysaccharide, LPS-challenge) from control pigs (sham-handled) using time point sampling. Using the same model, the second objective was to evaluate the sensitivity, specificity, AUC, and cutoff values of behavioral outputs collected continuously by the visually based precision technology. The authors hypothesized that human identification of challenged pigs would greatly decline over time, however, the precision technology would only moderately decline as pigs recover over time, and that human identification of pigs within a pen that were challenged with either LPS or sham-handled would have decreased sensitivity, specificity, and AUC than precision behavioral outcomes.

## 2. Materials and Methods

### 2.1. Animals and Housing

All experimental procedures adhered to the *Guide for the Care and Use of Agricultural Animals in Research and Teaching*. All procedures were reviewed and approved by the University of Nebraska—Lincoln Institutional Animal and Care and Use Committee (IACUC #1409).

One hundred and ninety-two newly weaned pigs (gilts and barrows) were sourced from the University of Nebraska—Lincoln Eastern Nebraska Research and Extension Center’s swine unit and housed in 12 nursery pens (16 pigs/pen) within one nursery room. Nursery pen flooring was tenderfoot flooring (Tandem Products, Inc. Minneapolis, MN, USA), with a 2.44 m^2^ solid mat (Rubber-Cal, Fountain Valley, CA, USA) placed at the front of the pen. The side and back walls were solid cement, and the front gate was made of vertical stainless steel bars (Farmweld, Teutopolis, IL, USA). The feeder was in the front of the pen and the waterer was placed at the back of the pen. The temperature in nursery rooms was 27.8 °C. Humidity ranged from 60–70%. Ad libitum water was provided through water nipples, and feed was offered through an 8-hole feeder. Diet was formulated to meet the NRC requirements for nursery pigs [23]. At the time of processing (0–2 days after birth), pigs had a generic button tag added to both ears. Pigs were weaned between 21–25 days of age.

### 2.2. Treatments after Weaning

At weaning, a unique color and alpha-numeric ear replaced the generic tag, which allowed the visual tracking system to autocorrect and recover lost individual identification [22]. A day before treatment (d − 1; Figure 1), all pigs were weighed to calculate the dosages of treatments appropriately.

This experiment was a portion of a larger project that included commingling or remaining with groupmates at the finishing stage (will be reported elsewhere). First, pigs were randomly assigned pens, stratifying gender (barrows and gilts) across 24 pens. Then, finishing pens were randomly assigned to commingling or noncommingling. Then, those pens were randomly assigned one of three nursery treatments (this report) so that nursery pigs began in pens of 16.

During the nursery phase, the 16 pigs were placed in 12 pens. Nine days after weaning, pigs were administered treatments. The nursery had 3 types of pens, 4 pens each: (1) In control pens, pigs were sham-handled (Sham; 3 mL of sterile saline); (2) In all-challenged pens, all 16 pigs were administered lipopolysaccharide (LPS; isolated from cell walls of heat-killed Escherichia coli O111:B4; Sigma-Aldrich, St. Louis, MO, USA) at a dose of 300 μg/kg body weight, dissolved in 3 mL of sterile saline, and; (3) In half-and-half pens, 8 pigs were administered LPS, and the other 8 were sham-treated. All injections were administered subcutaneously in the left medial inguinal area using a syringe with a ½ inch long 21-gauge needle. The IACUC leading veterinarian required rescue interventions for LPS-treated pigs that became completely unresponsive to external stimuli on the day of treatment. To improve cardio-respiratory function, the protocol included epinephrine (0.5 mL/pig) and dexamethasone (0.5 mL/pig) intramuscularly.

### 2.3. Precision Monitoring System

The NU*track* system’s cameras were installed at a 90-degree angle above the 12 nursery pens before the pigs’ placement. This deep learning-based, multiobject tracking system can achieve greater than 92.5% precision and recall when tracking individual pigs’ long-term individual identity, location, and individual posture in group-housed settings [15]. The hardware component of the system was FLIR/Lorex NVR System (Lorex Corporation, Linthicum, Maryland). The IP-based cameras had 4K (8 MP) and infrared capability for low-light recording. Visual data were collected continuously at a rate of 5 frames per second. Visual data were then pushed to a Dell-Alienware GPU-equipped desktop computer for processing. The software used fully convolutional networks to detect the location and orientation of pigs and their ear tags. The software aspect of the NU*track* system was based upon a Bayesian multiobject tracking approach, as reported previously [19,20,21,22]. This method combined the visual classification of ear tags with frame-to-frame movement probabilities, which allowed for correct identification, even when ear tags were obscured. Measures for this report (Figure 2) included the distance each pig traveled (m), angle (radius, pivot behavior), as well as the durations of stand, walk, and lie (lie-sternal, lie-lateral, sit).

### 2.4. Human-Derived Data

Live time point data were methodically collected by two expert human observers (veterinarian and trained technician) with interobserver reliability > 95%. Human observers were blinded to the treatments. After reliability was established, each observer stood in front of the pen and classified the animals according to a visual sickness (Figure 3) and hide score (Figure 4) for three minutes.

The sickness scores were adapted from established calf health scores [27]. The scores ranged from 0 to 2: normal and alert were scored as 0; sleepy or drowsy was scored as 1, and nonresponsive to stimuli was scored as 2. The score of 2 included lateral lying and open-mouth respiration. Hide scores (Figure 3) and emesis events were recorded but were not used in the final analyses since individual’s hidescould not be seenwhen pigs rested on top of each other (a.k.a. pig-pile) Emesis events only occurred on the day of the challenge and were not frequent enough to be included in the analyses.

### 2.5. Statistical Analysis

The behavioral data collected from the precision technology were analyzed using the receiver operator characteristic curve (ROC) analysis on RStudio (RStudio: Integrated Development for R. RStudio, PBC, Boston, MA, USA). The first analyses included the pens with all sham-treated pigs, all LPS-treated pigs, and half-and-half pigs. Then, the half-and-half pens were analyzed separately. From these analyses, the area under the curve (AUC) was measured to evaluate the predictive ability of the precision data, which statisticians consider a superior measure of accuracy. The AUC ranges from 0 to 1, an outcome of 0.5 means that the measure cannot discriminate between LPS and sham pigs, and a 1 means that the measure is perfect at identifying the treated animals (Figure 3). The human observation data were used as a contrast reference. Human data and precision data were fitted to a logistic regression equation, using the binomial family option to predict the known outcome of sick (received LPS) or not sick (did not receive LPS) individuals. ROC curves were plotted, and AUC values for the regression, pivot behavior, rest behavior, distance traveled, and human sick scores were calculated starting on treatment day (d − 0) and moving individually to each day post-treatment (d − 1 to d − 7). Both precision and human data were tested for the validity (sensitivity and specificity) of screening pens to find sickness in pigs as they recover from their treatments.

The precision data cutoff values were calculated on RStudio based on Youden’s index. The optimal cutoff point was then chosen by the maximum Youden index, maximizing the sum of sensitivity and specificity [28,29]. Optimal cutoff points were calculated for precision data only because human-derived data are considered categorical. In contrast, the precision data collected continuous variables based on the pigs’ behaviors. Once the optimal cutoffs were established, data could then be converted to a binomial categorical variable with two factors: LPS or sham treated. Pigs’ initial human-derived data and precision-derived data were included in the dataset for treatment day (d − 0).

## 3. Results

### 3.1. General Results

Hide scores were not included in the analyses since pigs rest and pile on top of each other, and human observers were not permitted to influence behavior unless there was an intervention event. Emesis events were also not included in the analyses since pigs ingested the vomit prior to data collection, and these records were infrequent. Nonetheless, a total of 24 events of emesis were recorded within the LPS-challenged pigs. Three of the 24 pigs that had recorded emesis became nonresponsive. Nine pigs received the rescue protocol intervention, however, those pigs did not recover and died from the endotoxin challenge, translating into a case fatality of 9.37%. For future work with LPS and unresponsive pigs, a euthanasia protocol should be used instead of a rescue protocol to improve animal welfare.

### 3.2. Human Data for Entire Population

When all pens were considered, human-derived data presented acceptable AUC, true positive rates, and true negative rates (0.85 AUC; >70% Sensitivity; >85% Specificity; Table 1) on days zero and one. By day two, human data AUC, true positive, and true negative rates declined to less desirable rates (Table 1). On days three and seven, human observations were not conducted because of extenuating circumstances. For days four, five, and six, AUC and true negatives were at less desirable rates (<0.57 AUC; <33% Specificity; Table 1), however, true positives remained acceptable (>83% Sensitivity; Table 1). However, true positive rates declined again for the last human observation time point, day six (Table 1).

The area under the curve (AUC) was calculated using the linear regression model and provides the validity measures for each behavior computed by NU*track*, with a zero to one range (0.5 = no discrimination; 1 = perfect performance); Optimal Cutoffs were based on maximum Youden’s index, maximizing the sum of sensitivity and specificity [28,29,30].

### 3.3. Precision Data for Entire Population

For the entire population of pigs, all precision measures had superior AUC values, true positive rates, and true negative rates compared to human data on days zero, one, and three (>0.98 AUC; >79% Sensitivity; >94% Specificity). The sensitivity value from human data was greater than distance and pivot-precision data for day four (84.4% vs. 72.9% and 80.2%, respectively). On day four, only lying duration (both positions) had a marginally acceptable AUC (0.715). Human data had greater sensitivity (84.4 %) than the precision data on day four (Table 1). Nonetheless, false positives were less frequent among the precision data for days four and five, compared with the human data (Table 1), especially for the lie duration (80.2% specificity for day five, Table 1). The area under the curve can serve as a reliable prediction analyses toolset, and all precision data behaviors had excellent AUC for precision data for days zero to two (>0.90; Table 1), except for the day four feed duration (0.787 AUC). However, precision data AUC declines steadily after day three.

### 3.4. Half LPS and Half Sham Data

When only the data from pens that had half of the pigs treated with LPS and the other half sham-treated were evaluated, human data had a lower AUC for all comparable precision data AUC (Table 1). However, the true negative rate for human scores was remarkably better on days four to six on half-and-half pens than when the entire population was scored (Table 1 and Table 2). The precision measures of pigs in the half-and-half pens resulted in AUCs that were comparable (> 0.99) to the entire population for days zero and one, however, the AUC declined starting on day two (Table 1 and Table 2). The sensitivity of precision data for the half-and-half pigs were greater than the comparable human-derived sensitivity (Table 2), with a >90% sensitivity on day one for the distance moved and the feeding duration. A noteworthy observation is that feeding duration sensitivity increased from 73.9% to 100% when the entire population vs. half-and-a-half pigs were observed (Table 1 and Table 2).

The area under the curve (AUC) was calculated using the linear regression model and provides the validity measures for each behavior computed by NU*track*, with a zero to one range (0.5 = no discrimination; 1 = perfect performance); Cutoffs were based on Youden’s index, maximizing the sum of sensitivity and specificity [28,29,30].

## 4. Discussion

The use of endotoxin challenge to replicate sickness without risking infection is a well-established controlled immune challenge in both human studies and animal models [31]. Lipopolysaccharide is harvested from the outer membrane of heat-killed gram-negative bacteria, and researchers can control the amount and location administered. The administration of greater doses can replicate a cytokine and febrile response that is comparable to septic shock and low doses can be used to replicate preclinical sickness [32]. In this experiment, LPS was used to create positive and negative sick subjects, and better understand a precision technology’s ability to identify these subjects whether all of them were challenged, not challenged, or mixed together. The aforementioned pen treatment would not be possible to control if a live pathogen was used.

One of the goals of precision livestock technology is to provide support to the workforce as a means to improve the health and welfare of livestock. Animal technicians in large swine operations do not have the time needed to observe animals continuously. Instead, they rely on previous experience and use timepoint sampling to identify compromised pigs from noncompromised pigs. The authors used a more methodical score system for live, time-point observations for the purpose of comparing the capabilities to a precision livestock technology. Humans and this technology both rely on visual cues from the pigs. The human compares the structural changes of compromised pigs to pen mates, whereas precision data rely on both individual animal structural changes and spatial relationships using temporal data. Behaviors that are valid in indicating illness are those with high sensitivity and high specificity when compared with a “golden standard” [33]. Measures of sensitivity and specificity have been previously used to assess the validity of precision livestock technology in diverse animal production systems [34,35,36]. The behaviors selected in this study were measured as continuous variables, so optimal cutoff values were necessary to maximize sensitivity and specificity [37].

The ability of the precision technology to correctly identify pigs based on their behavior reflects the pig’s physiology. In general, immune-compromised pigs will have a decreased intake of substrates (water and feed) and an increased resting time to conserve energy while the immunological insult is resolved [38].

The ability of a human to identify an immune-challenged pig on the day of challenge (d − 0) was adequate. However, when both the entire population and the half-and-half pigs were examined, precision technology had more days with adequate sensitivities and specificities. Human observations created more false positives and false negatives than precision measures when the entire population was considered. In commercial swine operations, false-positive pigs may be treated with antibiotics, which can impact the rate of antibiotic-resistant pathogens. A more likely challenge in production is false negative pigs. False negative pigs may go undetected, even by experienced technicians. These pigs can potentially serve as vectors, especially as they are commingled into the finishing phase.

Some behaviors can be tracked by humans (e.g., lie, and the spatial location at a time point), however, precision technology provides additional measures that the human cannot assess without technology. For example, pivot behavior is challenging to assess with human eyes [24,39]. Pivot behavior captured by the current technology had good sensitivity and specificity to detect challenged pigs, and an excellent AUC. Nordgreen et al. (2018) [40] studied the effects of a low-dose LPS challenge on the behavior and brain neurotransmitters in pigs. They harvested pigs 72 h after the challenge and found that neurotransmitters and markers of inflammation in the brain (e.g., hippocampus, hypothalamus, amygdala) were still elevated [41]. This same research group and others conducted follow-up studies with group-housed pigs where they reported that the most significant change in social behavior was an increase in LPS-challenged pigs ear biting their pen mates [42,43,44]. Ear biting is an active, social behavior, rather than a classic lethargic behavior, that would not be intuitive for animal technician identification of compromised pigs. This active state of social interaction may explain why human data had an unacceptable sensitivity and specificity just 48 h after the challenge, while precision data had acceptable sensitivity and specificity 72 h later for the current experiment. During this interaction, the pig that is being bitten will likely turn away (pivot) away from the offender, therefore more research is needed regarding pivot measures and social interactions after immune challenges.

The usage of precision livestock technology is not a substitute workforce for swine operations. Instead, it serves as an additional tool to help in decision making. The presence of caretakers to check pigs’ health is necessary since it causes a moderate disruption of behaviors, providing pigs with a chance to show exploratory behaviors that will facilitate the identification of sick individuals by precision technology. For example, in the current experiment, the human observations from the half-and-half pens had lower false positives than the entire population data, especially as pen mates recovered (i.e., days four to six). This finding was not surprising to the authors since previous experiments using continuous measures from repeated human approach tests indicated that compromised pigs will be less responsive and move at a slower pace when a human is standing in front of the pen [45,46]. In the pens that had a mix of compromised pigs and healthy pigs, technicians can directly compare each pig’s behavior with their pen mates. This finding indicates that a human timepoint sample, in conjunction with precision technology, may be more ideal than expecting precision technology to completely take away human observation. For example, a technician could spend a few minutes observing each pen, record the obvious sick pigs, and then precision technology can provide the granular measures of behaviors within a pen. Precision data provide granularity because optimal cutoff values can be extrapolated for maintenance and social behaviors. The AUC from precision data can serve as a better predictor for other endpoints such as performance, or risk of morbidity and mortality.

## 5. Conclusions

The precision technology has great AUC, sensitivity, and specificity when compared with human observations, especially during the first 72 h after observation. Sensitivity and specificity are expected to decrease as pigs recover, however, AUC may be an important measure for determining the risk of morbidity or the need for medical intervention later in the pig’s life. Further experiments are needed for determining the precision technology’s ability to detect compromised pigs among a greater number of healthy pigs. Nonetheless, this technology in combination with timepoint human observations may serve as an optimum system for the semi-real-time identification of sick pigs.

## 6. Patents

Psota, E. T., T. B. Schmidt, L. Perez, and B. Mote. 2022. Animal detection based on detection and association of parts. NUtrack Livestock Monitoring System. US Patent Number: 11393088. Publication Date: 2022/7/17. Psota, L. Perez, M. Mittek, and T.B. Schmidt. 2020. System for tracking individual animals in a group-housed environment NUtrack Livestock Monitoring System. US Patent Number: 16114565. Publication Date: 2020/10/3.

## Figures and Tables

**Figure 1 animals-13-00246-f001:**
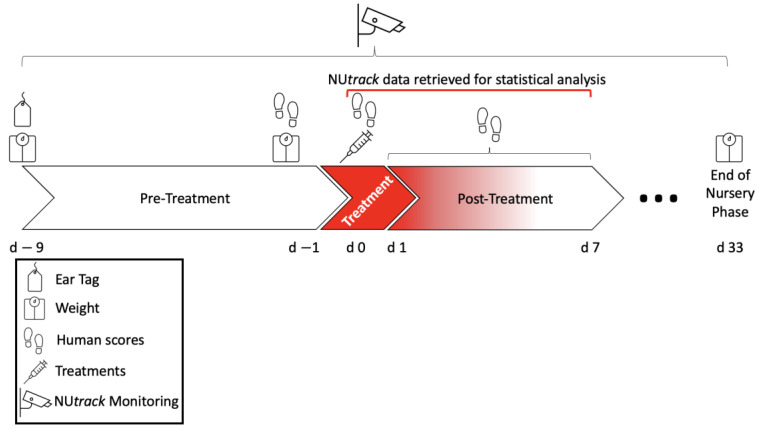
Timeline of experiment and data collection. Newly weaned pigs (n = 192) were randomly assigned to one of 12 pens (4 pens per treatment). Generic ear tags were replaced with NU*track* ear tags. Body weights were recorded on days 9 and 1 relative to the immune challenge (day 0). At 0900–0930 h on treatment day (d − 0), pigs received one of the treatments assigned to pens: (1) all sham-handled, controls (injection of 3 mL saline solution); (2) entire pen challenged with a single subcutaneous dose of lipopolysaccharide (LPS from E. coli O111:B; 300 μg/kg of body weight in a total of 3 mL saline), and; (3) one-half of the pigs were sham-handled, and one-half of the pigs were treated with the same amounts as described earlier (half-and-half). Human data were collected starting on days 1, 0, and once per day during seven days post-treatment (d − 1 to d − 7). A precision livestock technology (NU*track*) was utilized to continuously capture behavior measures (distance traveled, pivot behavior, feed, and total time lying) during the entirety of this timeline (d − 9 to d − 7). The 24-h data from the precision technology (NU*track*) data were used for this experiment.

**Figure 2 animals-13-00246-f002:**
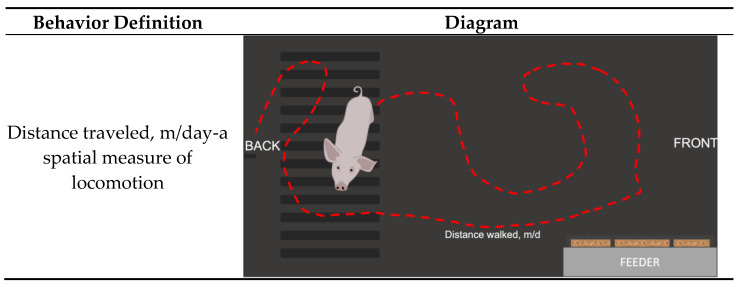
Ethogram used by the precision technology (NU*track*) to automatically track nursery pig behavioral data. ^1^ Pivot, rad/day- structural turning of head plus movement of front limbs while back limbs are still [24]; ^2^ Feed, min/day- Total duration overhead over feeder [25,26]; ^3^ Lie, min/day- total duration of Sternal or recumbent rest. The legs are legs straight, bent, or tucked under the pig’s body [22].

**Figure 3 animals-13-00246-f003:**
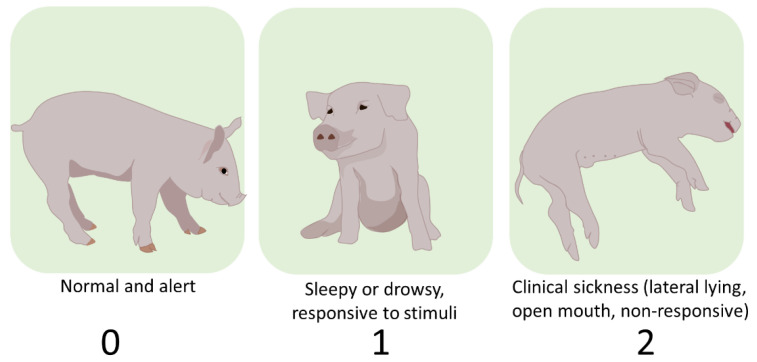
Human data: Posture and alertness categories and scores used by trained observers to classify pigs according to their clinical signs. Humans collected data for each pig in a pen, while standing in front of the pen for 3 min.

**Figure 4 animals-13-00246-f004:**
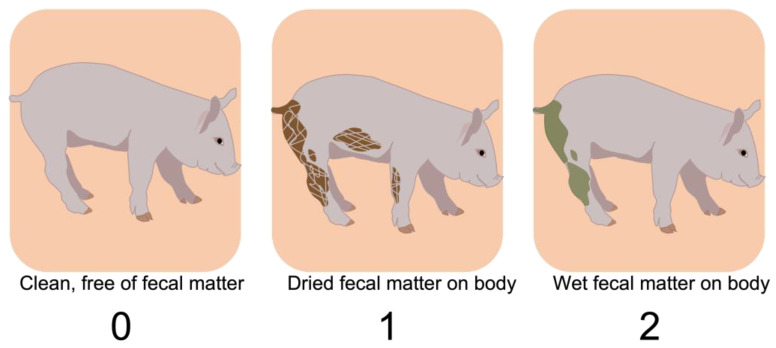
Human data- Hide score used by observers to classify pigs according to their clinical signs. In addition, emesis events were recorded. Humans collected data for each pig in a pen, while standing in front of the pen for 3 min.

**Table 1 animals-13-00246-t001:** The entire population of nursery pig optimal cutoff values, the area under the curve (AUC), sensitivity, and specificity for human data and precision data (NU*track*). Pigs received treatments randomly assigned by pen: (1) pens with all sham-handled, control pigs (injection of 3 mL saline solution); (2) pens with all challenged-pigs (subcutaneous dose of lipopolysaccharide (LPS from E. coli O111:B; 300 μg/kg of body weight in a total of 3 mL saline), or (3) pens with one-half of the pigs sham-handled, and the other half challenged ( same amounts of LPS as described earlier (i.e. half-and-half). Human real-time live scoring (human data) and data from a precision livestock technology (NU*track*; precision data) were evaluated.

	Day							
	0	1	2	3	4	5	6	7
Human Data								
*Sickness Score*								
Cutoff	-	-	-	-	-	-	-	-
AUC, 0–1	0.871	0.849	0.614	-	0.570	0.503	0.510	-
Sensitivity, %	88.5	70.7	53.4	-	84.4	83.3	68.8	-
Specificity, %	85.4	97.7	68.8	-	28.7	17.2	33.3	-
Precision Data								
*Distance traveled*								
Cutoff, m/day	784.6	365.7	958.7	1177.8	1132.1	1078.2	1008.4	1870.0
AUC, 0–1	0.981	0.999	0.929	0.846	0.686	0.542	0.481	0.391
Sensitivity, %	94.7	98.9	79.1	81.2	72.9	89.5	94.7	12.2
Specificity, %	91.6	98.8	96.2	76.7	60.9	21.8	12.6	93.1
*Pivot behavior*								
Cutoff, rad/d	2668	1384	3317	4151	3859	3761	4073	7583
AUC, 0–1	0.999	0.998	0.935	0.798	0.627	0.479	0.440	0.324
Sensitivity, %	96.8	98.9	82.2	83.3	80.2	86.4	73.3	10.0
Specificity, %	92.7	97.7	95.4	70.9	49.4	24.1	28.7	10.0
*Feed*								
Cutoff, s/d	7681	6885	10,922	9148	11,376	11,489	9029	18,136
AUC, 0–1	0.992	0.987	0.787	0.695	0.628	0.588	0.490	0.428
Sensitivity (%)	95.8	96.8	73.9	92.7	48.9	51.0	84.3	4.1
Specificity (%)	93.7	94.3	70.4	39.5	74.7	64.3	20.6	9.8
*Total lie*								
Cutoff, s/d	69,390	74,387	67,042	65,262	67,750	68,346	66,379	56,058
AUC, 0–1	0.993	0.996	0.903	0.800	0.715	0.518	0.487	0.381
Sensitivity, %	97.9	96.5	90.9	81.3	57.7	19.5	33.1	10.0
Specificity, %	94.8	97.9	77.0	69.7	80.2	88.5	75.0	20.8

**Table 2 animals-13-00246-t002:** Half-and-half nursery pig optimal cutoff values, the area under the curve (AUC), sensitivity, and specificity for human data and precision data (NU*track*). Four pens out of sixteen were randomly assigned the half-and-half treatment, then one-half of the pigs were randomly assigned treatments of either sham-handling, (controls; 3 mL saline solution subcutaneous) or single subcutaneous dose of lipopolysaccharide (LPS from E. coli O111:B; 300 μg/kg of body weight in a total of 3 mL saline). Human real-time live scoring (human data) and data from a precision livestock technology (NU*track* precision data) were evaluated.

	Day							
	0	1	2	3	4	5	6	7
Human Data								
*Sickness Score*								
Cutoff	-	-	-	-	-	-	-	-
AUC	0.703	0.738	0.569	-	0.662	0.525	0.513	-
Sensitivity, %	75.0	92.9	50.0	-	46.4	14.3	21.4	-
Specificity, %	65.6	53.1	62.5	-	84.4	90.6	81.2	-
Precision Data								
*Distance traveled*								
Cutoff, m/day	784.6	304.3	803.3	1191.1	1251.1	1449.5	1478.0	1578.7
AUC	0.970	0.992	0.799	0.729	0.710	0.680	0.648	0.580
Sensitivity, %	90.6	100.0	81.2	75.0	65.6	59.3	50.0	43.7
Specificity, %	96.8	96.4	75.0	67.8	75.0	82.1	78.5	78.5
*Pivot behavior*								
Cutoff, rad/d	2863	1489	2891	4610	4199	4578	4341	5572
AUC	0.988	0.985	0.792	0.676	0.663	0.629	0.612	0.517
Sensitivity, %	90.6	96.9	84.3	59.3	71.8	65.6	75.0	25.0
Specificity, %	100	92.8	75.0	82.1	57.1	67.8	50.0	89.2
*Feed*								
Cutoff, s/d	7856	5554	10,922	9358	8667	10,405	11,955	18,136
AUC	0.976	0.967	0.685	0.643	0.641	0.612	0.557	0.422
Sensitivity, %	87.5	100.0	71.8	93.7	90.6	81.2	53.1	62.5
Specificity, %	96.8	89.2	64.2	46.6	42.8	42.8	64.2	100.0
*Total lie*								
Cutoff, s/d	67,660	74,387	68,184	65,262	67,442	65,125	65,280	64,283
AUC	0.988	0.980	0.787	0.762	0.775	0.703	0.665	0.487
Sensitivity, %	100.0	89.2	75.0	78.5	67.8	64.2	60.7	50.0
Specificity, %	87.5	96.8	75.0	78.1	87.5	75.0	81.2	59.9

## Data Availability

Not applicable.

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
