# Peer review of "Evaluation of Precision Livestock Technology and Human Scoring of Nursery Pigs in a Controlled Immune Challenge Experiment"

_animals, 2023, doi:10.3390/ani13020246_

Round 1
Reviewer 1 Report
This paper discusses the important topic of utilisation of technology to support farmers in early detection of health compromises in growing pigs. Below are some comments on the manuscript which I feel should be addressed.
Abstract – include in the sensitivity / specificity numbers of the half and half pigs to go with the entire population numbers presented.
Introduction – You discuss the increased risk of morbidity / mortality following weaning – can you include some numbers to support this to showcase the scale of the problem / demonstrate the need to technology to support farmers at this time point
Methods 2.1 – were piglets assessed at weaning for any signs of injury / disease to prevent confounds in the randomly allocated study groups?
Methods 2.2 – what does ‘recovers lost individual identification’ mean? Does this relate with the system being able to begin tracking a pig again that has dropped off tracking?
Methods 2.2 – how may pigs required rescue intervention? This section is concerning. The criteria for this was ‘pigs that became completely unresponsive to external stimuli’ – this implies a significantly compromised welfare state – why was this decided as the point at which to treat and is this appropriate to wait to this stage? How frequently were the pigs monitored to prevent any pigs remaining in this severe state? Could the welfare of these animals have been improved?
Methods 2.4 – can you provide numbers on emesis events? Even if not used in analysis, this is indicating a compromise to welfare – was it these pigs that then went on to require intervention? Could this have been used as an earlier indicator of a significant problem and used as a marker for earlier rescue intervention?
Section 3.1 – this fatality rate is very high. How does it compare to the rate in the standard pig population on this particular farm? How does the rate link with pigs that did or did not have rescue intervention? Can you please provide information on how the pigs were killed.
Human observations – the data recorded in human observation and by NUtrack are not the same. When the human observations have recorded e.g. false positives (that the NUtrack system has not), could it be that the humans are observing signs of compromised health and welfare linked to another problem, that the NUtrack system would not detect, based on the parameters it is trained to recognise? Hence it appears like it is better – but actually could the humans observation be picking up something additional?
Linked to the point above – were any pigs excluded from the analysis based on health / welfare compromises thought not to be linked to the experimental trial? Was there any assessment of this?
This sort of technololgy could improve early detection of health compromises, but a significant barrier to implementation of technology on farm is the initial cost associated with purchase & installation. Can you comment on how this sort of system would work practically for a standard farmer?
Author Response
December 08, 2022
Dear Reviewers and Editor,
On behalf of my coauthors, we would like to thank you for the revisions and the opportunity to resubmission of our manuscript animals-2047254 entitled “Evaluation of Precision Livestock Technology and Human Scoring of Nursery Pigs in A Controlled Immune Challenge Experiment.” The comments and suggestions made by the reviewers were extremely helpful and we have considered and addressed each suggestion. Most of the suggestions were successfully incorporated into our manuscript.
We included a response to each reviewer’s suggestions. Questions from reviewer number 1 are signalized as R1, reviewer number 2 as R2, and the authors' responses as AU. Corresponding changes in the manuscript are marked with track changes using Microsoft word.
Sincerely,
Lindsey E. Hulbert, PhD, Associated Professor and Eduardo Mazzardo Bortoluzzi, DVM, PhD; Post Doctoral scholar
Kansas State University
127 Call Hall, Manhattan, KS 66506
785-532-0938
lhulbert@ksu.edu
Reviewer 2
The study confronts the problems of intensive swine industries and has specific research and application value. Although using interdisciplinary knowledge to solve pig farm problems will be an essential research tool in the future, it also faces significant challenges. There are many problems with applying intelligent technology in the production process, particularly in data collection, which is susceptible to many environmental factors that increase the difficulty of the collection techniques and, thus, the accuracy of the data collected. In this study, many environmental factors interfered with, for example, the piglets' social environment, temperature, and group behaviour. Therefore, the study needs further work.
R2: The author, Hui Hu, needs to include information on his affiliation.
AU: Affiliations for Hui Wu have been corrected. Thank you for pointing that out.
R2: The article's description of the material methods need to be more detailed. The author should describe in detail information such as temperature, humidity, flooring material, and type of flooring.
AU: The following paragraph was added to the materials and methods section to address nursery room environment:
“Nursery pen flooring was tenderfoot flooring (Tandem Products, Inc. Minneapolis, MN, USA), with a 2.44 m2 solid mat (Rubber-Cal, Fountain Valley, CA, USA) placed at the front of the pen. The side and back walls were solid cement, and the front gate was vertical stainless steel bars (Farmweld, Teutopolis, IL). The feeder was in the front of the pen, the waterer was placed at the back of the pen. The temperature in nursery rooms are 27.8 ⁰C. Humidity ranged from 60-70%.”
R2: In part 2.3, there is no problem with the technical approach of these sophisticated monitoring systems, which can determine the various postures of pigs with this deep learning-based multi-objective tracking system. However, the accuracy of the discrimination is affected when the pigs gather (pile up). In addition, the judgment's outcome is related to the environment and feeding conditions. How can we ensure the collected behavior is triggered by the LPS injection?
AU: While we cannot replicate every type of housing system design for nursery pigs, we can design a controlled, experimental study with controls and randomization. This study has two types of controls: Sham pigs (just given saline) that are all housed with only other Sham pigs, and Sham pigs that are housed with LPS pigs. The pens are identical, and we randomly assigned treatments to pens, and in the case of the half-and-half pens, once that treatment was randomly assigned, each pig was randomly assigned sham or LPS. Our math/computer engineer/statistical expert (Dr. Jaberi, co-author) wrote the program for randomization and approved the experimental design.
Furthermore, LPS challenge projects have been extensively reported in immunological and behavior studies for several decades. The inflammation caused by the endotoxin challenge affects animals’ behavior budgets, especially by decreasing activity, feeding time, and increasing lying time. Behavioral changes induced by endotoxin challenge inflammation were reported by many scientists studying pigs (Veit et al., 2021; Nordgreen et al., 2018; Lay et al., 2011; Johnson and Borell, 1994). Furthermore, a pilot study was done prior to this study to ensure we were achieving behavior modifications with the endotoxin challenge.
Veit, Christina, Andrew M. Janczak, Birgit Ranheim, Judit Vas, Anna Valros, Dale A. Sandercock, Petteri Piepponen, Daniela Dulgheriu, and Janicke Nordgreen. "The effect of LPS and ketoprofen on cytokines, brain monoamines, and social behavior in group-housed pigs." Frontiers in veterinary science7 (2021): 617634.
Nordgreen, Janicke, Camilla Munsterhjelm, Frida Aae, Anastasija Popova, Preben Boysen, Birgit Ranheim, Mari Heinonen et al. "The effect of lipopolysaccharide (LPS) on inflammatory markers in blood and brain and on behavior in individually-housed pigs." Physiology & behavior 195 (2018): 98-111.
Lay Jr, D. C., H. G. Kattesh, J. E. Cunnick, M. J. Daniels, G. Kranendonk, K. A. McMunn, M. J. Toscano, and M. P. Roberts. "Effect of prenatal stress on subsequent response to mixing stress and a lipopolysaccharide challenge in pigs." Journal of animal science 89, no. 6 (2011): 1787-1794.
Johnson, Rodney W., and E. Von Borell. "Lipopolysaccharide-induced sickness behavior in pigs is inhibited by pretreatment with indomethacin." Journal of animal science 72, no. 2 (1994): 309-314.
R2: How can we address the individual differences in nursery pigs when faced with different disease risks? For example, some pigs are infected with contagious diseases but do not produce corresponding behavioural changes. It is recommended that multiple factors be combined to determine this rather than a single video capture.
AU: As stated previously, we created an experimental design with controls and randomization. You are correct that a live pathogen would be complicated and not guarantee that all pigs are sick. LPS not live and is not contagious, which is why we chose it as a controlled in vivo immune challenge, rather than a contagious disease. The senior authors have a lot of experience with LPS and with live immune challenges.
R2: The data collected in this study cannot be fully identified as abnormal behaviour in an inflammatory state. However, they may also be related to other environmental factors, and therefore a large sample size is needed to improve precision.
AU: As stated previously, LPS is a very well-studied endotoxin that creates a very proinflammatory state without making the animal contagious. LPS is commercially made for in vitro, ex vivo, and in vivo projects. The producers first take gram-negative bacteria, and first heat kill the bacteria. When this occurs, the cell is broken apart, and then the LPS on the cell wall of the bacteria is extracted. Across species, the immune systems have Toll-like receptors on the immune cells that bind to the natural produced LPS of dead and dying bacteria. They have evolved because dead and dying bacteria would most likely come from a compromised G.I. (e.g., the tight junctions are damaged). When LPS is recognized by circulating and resident leukocytes, they send a cascade of cytokines that are extremely pro-inflammatory. The most traditional use of LPS for humans back in the 1980’s was to replicate septicemia. It has now been used as a toolset for many species, especially for scientists who do not have access to BSL-2 or 3 facilities and want the animals to have little variation in responses, and similar, predictable recovery times. The LPS used in our project ensures that the animal is truly sick, which is why we can calculate sensitivity, even if the main effects last for a few days.
Additional Modifications to the Manuscript
Page 1 - Authors listing: Removal of an author that was not in our submission (Trenhaile-Grannemann, M.D.).
Page 1 - Keywords: 3 keywords were added to match submission keywords.
All pages – References and Citations: Adjustment of citation numbers due to the addition of citations to the manuscript.
Page 7,8,9 – Tables 1 and 2: adjusted table boarders to maintain consistency and adjusted “Precision Data” title on table 2 to its correct location.
Minor spelling and punctuation corrections are pointed out through the manuscript using track changes from Microsoft Word.
Reviewer 2 Report
The study confronts the problems of intensive swine industries and has specific research and application value. Although using interdisciplinary knowledge to solve pig farm problems will be an essential research tool in the future, it also faces significant challenges. There are many problems with applying intelligent technology in the production process, particularly in data collection, which is susceptible to many environmental factors that increase the difficulty of the collection techniques and, thus, the accuracy of the data collected. In this study, many environmental factors interfered with, for example, the piglets' social environment, temperature, and group behaviour. Therefore, the study needs further work.

Author Response
December 08, 2022
Dear Reviewers and Editor,
On behalf of my coauthors, we would like to thank you for the revisions and the opportunity to resubmission of our manuscript animals-2047254 entitled “Evaluation of Precision Livestock Technology and Human Scoring of Nursery Pigs in A Controlled Immune Challenge Experiment.” The comments and suggestions made by the reviewers were extremely helpful and we have considered and addressed each suggestion. Most of the suggestions were successfully incorporated into our manuscript.
We included a response to each reviewer’s suggestions. Questions from reviewer number 1 are signalized as R1, reviewer number 2 as R2, and the authors' responses as AU. Corresponding changes in the manuscript are marked with track changes using Microsoft word.
Sincerely,
Lindsey E. Hulbert, PhD, Associated Professor and Eduardo Mazzardo Bortoluzzi, DVM, PhD; Post Doctoral scholar
Kansas State University
127 Call Hall, Manhattan, KS 66506
785-532-0938
lhulbert@ksu.edu
Reviewer 2
The study confronts the problems of intensive swine industries and has specific research and application value. Although using interdisciplinary knowledge to solve pig farm problems will be an essential research tool in the future, it also faces significant challenges. There are many problems with applying intelligent technology in the production process, particularly in data collection, which is susceptible to many environmental factors that increase the difficulty of the collection techniques and, thus, the accuracy of the data collected. In this study, many environmental factors interfered with, for example, the piglets' social environment, temperature, and group behaviour. Therefore, the study needs further work.
R2: The author, Hui Hu, needs to include information on his affiliation.
AU: Affiliations for Hui Wu have been corrected. Thank you for pointing that out.
R2: The article's description of the material methods need to be more detailed. The author should describe in detail information such as temperature, humidity, flooring material, and type of flooring.
AU: The following paragraph was added to the materials and methods section to address nursery room environment:
“Nursery pen flooring was tenderfoot flooring (Tandem Products, Inc. Minneapolis, MN, USA), with a 2.44 m2 solid mat (Rubber-Cal, Fountain Valley, CA, USA) placed at the front of the pen. The side and back walls were solid cement, and the front gate was vertical stainless steel bars (Farmweld, Teutopolis, IL). The feeder was in the front of the pen, the waterer was placed at the back of the pen. The temperature in nursery rooms are 27.8 ⁰C. Humidity ranged from 60-70%.”
R2: In part 2.3, there is no problem with the technical approach of these sophisticated monitoring systems, which can determine the various postures of pigs with this deep learning-based multi-objective tracking system. However, the accuracy of the discrimination is affected when the pigs gather (pile up). In addition, the judgment's outcome is related to the environment and feeding conditions. How can we ensure the collected behavior is triggered by the LPS injection?
AU: While we cannot replicate every type of housing system design for nursery pigs, we can design a controlled, experimental study with controls and randomization. This study has two types of controls: Sham pigs (just given saline) that are all housed with only other Sham pigs, and Sham pigs that are housed with LPS pigs. The pens are identical, and we randomly assigned treatments to pens, and in the case of the half-and-half pens, once that treatment was randomly assigned, each pig was randomly assigned sham or LPS. Our math/computer engineer/statistical expert (Dr. Jaberi, co-author) wrote the program for randomization and approved the experimental design.
Furthermore, LPS challenge projects have been extensively reported in immunological and behavior studies for several decades. The inflammation caused by the endotoxin challenge affects animals’ behavior budgets, especially by decreasing activity, feeding time, and increasing lying time. Behavioral changes induced by endotoxin challenge inflammation were reported by many scientists studying pigs (Veit et al., 2021; Nordgreen et al., 2018; Lay et al., 2011; Johnson and Borell, 1994). Furthermore, a pilot study was done prior to this study to ensure we were achieving behavior modifications with the endotoxin challenge.
Veit, Christina, Andrew M. Janczak, Birgit Ranheim, Judit Vas, Anna Valros, Dale A. Sandercock, Petteri Piepponen, Daniela Dulgheriu, and Janicke Nordgreen. "The effect of LPS and ketoprofen on cytokines, brain monoamines, and social behavior in group-housed pigs." Frontiers in veterinary science7 (2021): 617634.
Nordgreen, Janicke, Camilla Munsterhjelm, Frida Aae, Anastasija Popova, Preben Boysen, Birgit Ranheim, Mari Heinonen et al. "The effect of lipopolysaccharide (LPS) on inflammatory markers in blood and brain and on behavior in individually-housed pigs." Physiology & behavior 195 (2018): 98-111.
Lay Jr, D. C., H. G. Kattesh, J. E. Cunnick, M. J. Daniels, G. Kranendonk, K. A. McMunn, M. J. Toscano, and M. P. Roberts. "Effect of prenatal stress on subsequent response to mixing stress and a lipopolysaccharide challenge in pigs." Journal of animal science 89, no. 6 (2011): 1787-1794.
Johnson, Rodney W., and E. Von Borell. "Lipopolysaccharide-induced sickness behavior in pigs is inhibited by pretreatment with indomethacin." Journal of animal science 72, no. 2 (1994): 309-314.
R2: How can we address the individual differences in nursery pigs when faced with different disease risks? For example, some pigs are infected with contagious diseases but do not produce corresponding behavioural changes. It is recommended that multiple factors be combined to determine this rather than a single video capture.
AU: As stated previously, we created an experimental design with controls and randomization. You are correct that a live pathogen would be complicated and not guarantee that all pigs are sick. LPS is not live and is not contagious, which is why we chose it as a controlled in vivo immune challenge, rather than a contagious disease. The senior authors experts in inflammation in livestock species and have 20+ years with using LPS in vivo and ex vivo in multiple species and understanding the challenges with evaluating behaviors with live, contagious immune challenges.
R2: The data collected in this study cannot be fully identified as abnormal behaviour in an inflammatory state. However, they may also be related to other environmental factors, and therefore a large sample size is needed to improve precision.
AU: As stated previously, LPS is a very well-studied endotoxin that creates a very proinflammatory state without making the animal contagious. LPS is commercially made for in vitro, ex vivo, and in vivo projects. The producers first take gram-negative bacteria, and first heat kill the bacteria. When this occurs, the cell is broken apart, and then the LPS on the cell wall of the bacteria is extracted. Across species, the immune systems have Toll-like receptors on the immune cells that bind to the natural produced LPS of dead and dying bacteria. They have evolved because dead and dying bacteria would most likely come from a compromised G.I. (e.g., the tight junctions are damaged). When LPS is recognized by circulating and resident leukocytes, they send a cascade of cytokines that are extremely pro-inflammatory. The most traditional use of LPS for humans back in the 1980’s was to replicate septicemia. It has now been used as a toolset for many species, especially for scientists who do not have access to BSL-2 or 3 facilities and want the animals to have little variation in responses, and similar, predictable recovery times. The LPS used in our project ensures that the animal is truly sick, which is why we can calculate sensitivity, even if the main effects last for a few days.
Additional Modifications to the Manuscript
Page 1 - Authors listing: Removal of an author that was not in our submission (Trenhaile-Grannemann, M.D.).
Page 1 - Keywords: 3 keywords were added to match submission keywords.
All pages – References and Citations: Adjustment of citation numbers due to the addition of citations to the manuscript.
Page 7,8,9 – Tables 1 and 2: adjusted table boarders to maintain consistency and adjusted “Precision Data” title on table 2 to its correct location.
Minor spelling and punctuation corrections are pointed out through the manuscript using track changes from Microsoft
Round 2
Reviewer 2 Report
agree